# Data Fusion of Dual Foot-Mounted INS Based on Human Step Length Model

**DOI:** 10.3390/s24041073

**Published:** 2024-02-07

**Authors:** Jianqiang Chen, Gang Liu, Meifeng Guo

**Affiliations:** 1Department of Precision Instrument, Tsinghua University, Beijing 100084, China; chenjq16@g.ecc.u-tokyo.ac.jp (J.C.); guomf@tsinghua.edu.cn (M.G.); 2Department of Electronic Engineering, Tsinghua University, Beijing 100084, China

**Keywords:** pedestrian navigation, MIMU, INS, ZUPT, human step length model, bipedal constraint algorithm

## Abstract

Pedestrian navigation methods based on inertial sensors are commonly used to solve navigation and positioning problems when satellite signals are unavailable. To address the issue of heading angle errors accumulating over time in pedestrian navigation systems that rely solely on the Zero Velocity Update (ZUPT) algorithm, it is feasible to use the pedestrian’s motion constraints to constrain the errors. Firstly, a human step length model is built using human kinematic data collected by the motion capture system. Secondly, we propose the bipedal constraint algorithm based on the established human step length model. Real field experiments demonstrate that, by introducing the bipedal constraint algorithm, the mean biped radial errors of the experiments are reduced by 68.16% and 50.61%, respectively. The experimental results show that the proposed algorithm effectively reduces the radial error of the navigation results and improves the accuracy of the navigation.

## 1. Introduction

The Global Navigation Satellite System (GNSS) is the most common navigation solution in an open outdoor environment, with an accuracy of up to a meter or even higher [1]. However, in an indoor environment or in a special environment where satellite signals cannot be received outdoors, the above satellite-based navigation solution cannot meet people’s needs [2]. Therefore, autonomous navigation solutions that rely only on their own sensors are needed to achieve navigation and positioning in a satellite-denied environment. Autonomous navigation systems play a crucial role in various scenarios, particularly in meeting the safety needs of workers in special situations such as tunnel operations and underground mines, assisting firefighters in rescue work at fire scenes, helping individuals navigate through the wild jungle, and providing guidance in large buildings such as shopping malls, libraries, and museums.

Pedestrian navigation systems are categorized into non-autonomous modes, primarily based on satellite navigation, and autonomous modes, which rely on sensors like the Inertial Measurement Unit (IMU) [3]. Different from the widespread use of GNSS in outdoor environments, indoor navigation and positioning uses more diverse methods. Pascacio et al. [4] conducted a survey and analysis of the main technical means used in indoor navigation and positioning. The proportions of each main technology and their related papers are as follows: Wireless Local Area Network (WLAN) accounts for 53.5%, IMU accounts for 30.9%, Ultra Wideband (UWB) accounts for 15.4%, Bluetooth (Bluetooth) accounts for 2.3%, 5G accounts for 2.3%, Radio Frequency Identification (Radio Frequency Identification, RFID) accounts for 2.3%, and vision cameras account for 1.1%.

Research has been conducted on pedestrian navigation technology since the 1980s. Currently, it is divided into two categories: Pedestrian Dead Reckoning (PDR) and Inertial Navigation System (INS).

The PDR algorithm can be deployed on different parts of the human body, such as the chest, waist, and feet, and has the advantage of easy installation and application [5]. Marth et al. first designed a personal navigation system composed of a pedestrian dead reckoning module and GPS in 1998. When satellite signals are denied, the system can only use accelerometers and magnetometers for dead reckoning [6]. Yao et al. proposed a more robust method to solve the difficulty of calculating the number of steps for gait detection caused by the traditional PDR in which the person is in the in situ traveling state, the core idea of the method is to recognize the walking patterns of pedestrians, such as normal walking and fast walking [7]. Basso et al. proposed to process vertical acceleration using a phase-locked loop to detect the stride length and number of steps, and then to calculate the step length using a segmented linear relationship with the stride length, as well as estimating the direction of the stride based on lumbar kinematics using geometric features of planar acceleration [8]. Based on the PDR algorithm, Liu et al. proposed a new indoor positioning solution by combining extended Kalman filtering with the Wi-Fi signal round-trip time, and the average positioning error was reduced by 37.4∼67.6% [9].

With the development of deep learning technology, many scholars try to use it to solve the old problems of the traditional PDR algorithm. Edle et al. applied BLSTM-RNN to gait detection to improve the stability and recognition ability of the algorithm [10]. Lin et al. [11] and Luo et al. [12] analyzed the pedestrian directions of forward, backward, left, and right using LSTM and CNN. Wang et al. designed a step estimation network based on LSTM, with a stride error rate of 4.63% and a walking distance error rate of 1.43% [13]. Furthermore, Asraf et al. designed PDRNet [14] based on ResNet, and Klein et al. designed StepNet [15,16], which directly uses accelerometer and gyroscope measurement data to output the direction and distance through the neural network. The steps of using accelerometer and gyroscope data to perform gait detection, and then completing step length estimation and heading estimation using the PDR method, are skipped. However, the above-mentioned PDR methods based on deep learning inevitably have the problem of a large computational workload and often require the use of computer graphics cards for calculations. Therefore, under the current circumstances, this solution is still mainly limited to the scope of laboratory research, which is not conducive to the embedded design of pedestrian navigation equipment and cannot be deployed in the short term.

With the continuous improvement of Micro Inertial Measurement Unit (MIMU) accuracy, many researchers have focused their research on INS-based pedestrian navigation solutions. Elwell et al., in 1999, proposed the idea of designing a Zero Velocity Update algorithm using pedestrian step characteristics [17]. Based on this, Foxlin designed the first foot-mounted pedestrian inertial navigation system in 2005 [18]. Borenstein et al. of the University of Michigan proposed the Heuristic Drift Reduction (HDR) method for estimating and eliminating the heading angle drift error [19]. Skog et al. of the Royal Institute of Technology, Sweden, quantitatively analyzed a variety of gait detection algorithms and ZUPT algorithms [20]. They demonstrated that gyroscope-based detection outperforms accelerometer-based detection under specific assumptions. Zhang et al. realized the combined pattern recognition of four gait patterns and four device poses [21]. Meanwhile, Xu et al. investigated an indoor localization algorithm combining IMU and UWB [22]. Garcia et al. proposed a new FootSLAM algorithm using only IMU data, showing the possibility of using human odometers for indoor mapping and precise navigation [23]. Zhang et al. also proposed a yaw angle error self-observation algorithm (YESO) to reduce the yaw angle error and calibrate the yaw angle [24].

Meanwhile, multi-IMU information fusion algorithms have gradually become a hot research topic in recent years. Qian et al. used the IMU of the trunk and lower limbs to form a virtual foot IMU which, in turn, realizes the pedestrian navigation function, and the final positioning accuracy is about 6% of the travel distance [25]. Based on a zero-speed detector based on hypothesis testing and great likelihood estimation designed using human gait information, Shi Wei et al. designed a Kalman-filtered ellipsoid-constrained algorithm under the maximal step length constraint while walking [26,27]. Xu Yuan et al., on the other hand, solved the problem of poor viewability of heading information in the traditional foot-based heading reference system by designing an indoor personal navigation system combining a foot-based heading reference system and a shoulder-based electronic compass [28,29]. Patel et al. conducted experimental analysis on four different fusion methods of multi-IMU, proving that the navigation accuracy of using five IMU fusion filters has been improved [30]. Niu et al. [31,32], Li et al. [33], and Prateek et al. [34] also studied pedestrian navigation algorithms based on dual foot-mounted IMU. They showed that, by limiting the distance between the two feet, the Kalman filter can be used to constrain the navigation error and obtain better robustness in the system.

However, the above studies did not make full use of human kinematic characteristics to suppress heading angle errors to improve navigation accuracy. In this paper, we built a human step length model using human kinematic data collected by the motion capture system. And based on the established human step length model, we propose the bipedal constraint algorithm to address the issue of heading angle errors accumulating over time in pedestrian navigation systems. The contributions of the paper are as follows:We collected human kinematics data using an optical–inertial fusion motion capture system. A whole-body musculoskeletal model of the human body was created to analyse navigation-related human kinematic parameters and build the human step length model. This provides a parameter basis for the navigation algorithm.We developed an embedded pedestrian navigation system that includes an IMU and a barometer. The system acquires and wirelessly transmits navigation data. We used this system to verify the effectiveness of the navigation algorithm.We proposed the bipedal constraint algorithm to address the issue of the ZUPT algorithm’s inability to constrain the heading angle. The virtual observation quantities and corresponding observation matrices of both the ZUPT algorithm and the bipedal constraint algorithm were constructed for application to the Kalman filter. The experiment demonstrates that the bipedal constraint algorithm effectively limits the heading angle error during navigation and enhances the accuracy of navigation and positioning.

The rest of the paper is organized as follows: in Section 2, we present the human step length model and the bipedal constraint algorithm. In Section 3, we describe the experiment in detail and evaluate the results. In Section 4, we provide a conclusion for our study and talk about future work.

## 2. Pedestrian Navigation System

The pedestrian navigation system is based on the Strapdown Inertial Navigation System (SINS) and Kalman filter (KF), and uses distance for coupling.

### 2.1. Strapdown Inertial Navigation System

The core of SINS is to use numerical integration to calculate the attitude, speed, and position of the carrier. Therefore, the SINS algorithm mainly includes three parts: an attitude update algorithm, a speed update algorithm, and a position update algorithm.

#### 2.1.1. Attitude Update Algorithm

The differential equation for the attitude of the carrier in the n-system of geographic coordinates is given by
(1)$${\dot{C}}_{b}^{n}=C_{b}^{n}\left(\omega_{nb}^{b}\times \right)$$

Here, $C_{b}^{n}$ is the attitude array of the b-system with respect to the n-system. $\omega_{nb}^{b}$ is the angular velocity of the b-system with respect to the n-system. Also, according to the law of chain multiplication of matrices, there is
(2)$$C_{b(m)}^{n(m)}=C_{i}^{n(m)}C_{b(m)}^{i}=C_{n(m-1)}^{n(m)}C_{b(m-1)}^{n(m-1)}C_{b(m)}^{b(m-1)}$$

$C_{b(m-1)}^{n(m-1)}$ and $C_{b(m)}^{n(m)}$ represent the attitude matrices at $t_{m-1}$ and $t_{m}$, respectively. $C_{b(m)}^{b(m-1)}$ represents the rotational change of the b-system from moment $t_{m-1}$ to moment $t_{m}$ using the i-coordinate system as the reference frame. $C_{n(m-1)}^{n(m)}$ represents the rotational change of the n-system from moment $t_{m}$ to moment $t_{m-1}$ using the i-coordinate system as the reference frame. When using the two-subsample cone error compensation algorithm, for $C_{n(m-1)}^{n(m)}$ and $C_{b(m)}^{b(m-1)}$, there is
(3)$$C_{n(m-1)}^{n(m)}=M_{RV}^{T}\left(\phi_{in(m)}^{n}\right)\approx M_{RV}^{T}\left(T\omega_{in(m)}^{n}\right)$$
(4)$$C_{b(m)}^{b(m-1)}=M_{RV}\left(\phi_{ib(m)}^{b}\right)$$
(5)$$\phi_{ib(m)}^{b}=\left(\Delta \theta_{m1}+\Delta \theta_{m2}\right)+\frac{2}{3}\Delta \theta_{m1}\times \Delta \theta_{m2}$$
(6)$$M_{RV}\left(\phi \right)=C_{b}^{i}=I+\frac{\sin\phi }{\phi }\left(\phi \times \right)+\frac{\left(1-\cos\phi \right)}{\phi^{2}}{\left(\phi \times \right)}^{2}$$

Here, $T=t_{m}-t_{m-1}$, $\omega_{in(m)}^{n}$ is the angular velocity of the n-system relative to the i-system at this time, $\Delta \theta_{m1}$ and $\Delta \theta_{m2}$ are the two angular increments in the period from $t_{m-1}$ to $t_{m}$, and ϕ represents the equipment rotation vector.

#### 2.1.2. Speed Update Algorithm

There is an inertial specific force equation in the n-system as follows:(7)$${\dot{v}}_{en}^{n}=C_{b}^{n}f_{sf}^{b}-\left(2\omega_{ie}^{n}+\omega_{en}^{n}\right)\times v_{en}^{n}+g^{n}$$

Here, $f_{sf}^{b}$ is the accelerometer output specific force, $2\omega_{ie}^{n}\times v_{en}^{n}$ is the Göttinger’s acceleration, $\omega_{en}^{n}\times v_{en}^{n}$ is the centripetal acceleration due to the motion of the carrier around the center of the earth, and $g^{n}$ is the acceleration due to gravity. Integration leads to the following formula:(8)$$v_{m}^{n(m)}=v_{m-1}^{n(m-1)}+\Delta v_{sf(m)}^{n}+\Delta v_{cor/g(m)}^{n}$$

Here, $v_{m-1}^{n(m-1)}$ and $v_{m}^{n(m)}$ are the velocities of the carrier at the moments $t_{m-1}$ and $t_{m}$, respectively, and there are
(9)$$\Delta v_{sf(m)}^{n}=\int_{t_{m-1}}^{t_{m}}C_{b}^{n}\left(t\right)f_{sf}^{b}\left(t\right)dx$$
(10)$$\Delta v_{cor/g(m)}^{n}=\int_{t_{m-1}}^{t_{m}}\left\{-\left[2\omega_{ie}^{n}\left(t\right)+\omega_{en}^{n}\left(t\right)\right]\times v_{en}^{n}\left(t\right)+g^{n}\left(t\right)\right\}dt$$

The numerical integration of Equations (Equation 9) and (Equation 10) can be approximated as follows:(11)$$\begin{array}{cc} \Delta v_{sf(m)}^{n}\approx & \left[I-\frac{T}{2}\left(\omega_{in(m-1/2)}^{n}\times \right)\right]C_{b(m-1)}^{n(m-1)}\\ \; & \left[\Delta v_{m}+\frac{1}{2}\Delta \theta_{m}\times \Delta v_{m}+\frac{2}{3}\left(\Delta \theta_{m1}\times \Delta v_{m2}+\Delta v_{m1}\times \Delta \theta_{m2}\right)\right] \end{array}$$
(12)$$\begin{array}{c} \Delta v_{cor/g(m)}^{n}\approx \left\{-\left[2\omega_{ie(m-1/2)}^{n}+\omega_{en(m-1/2)}^{n}\right]\times v_{en(m-1/2)}^{n}+g_{m-1/2}^{n}\right\}T \end{array}$$

Here, Δv is the specific velocity increment and Δθ is the angular increment.

#### 2.1.3. Position Update Algorithm

The matrix form of the differential equation for the position of the inertial navigation system is as follows:(13)$$\dot{p}=M_{pv}v^{n}$$

Here,
(14)$$p={\left[\begin{array}{ccc} L & \lambda & h \end{array}\right]}^{T}$$
(15)$$M_{pv}=\left[\begin{array}{ccc} 0 & 1/(R_{M}+h) & 0\\ \sec L/(R_{N}+h) & 0 & 0\\ 0 & 0 & 1 \end{array}\right]$$

Here, *L*, λ, and *h* represent the longitude, latitude, and altitude. $R_{M}$ represents the principal radii of curvature along the meridional section. $R_{N}$ represents the principal radii of curvature along the prime–vertical normal section. The numerical integration of Equation (Equation 13) yielded
(16)$$p_{m}=p_{m-1}+M_{pv(m-1/2)}\left(v_{m-1}^{n}+v_{m}^{n}\right)\frac{T}{2}$$

#### 2.1.4. Attitude Initialization

When the IMU is stationary, the equation for specific force can be simplified to
(17)$$0=C_{b}^{n}f_{sf}^{b}+g^{n}$$

Transform and expand Equation (Equation 17) into the component form:(18)$$\left[\begin{array}{c} f_{sfx}^{b}\\ f_{sfy}^{b}\\ f_{sfz}^{b} \end{array}\right]=-\left[\begin{array}{ccc} C_{1}^{T} & C_{2}^{T} & C_{3}^{T} \end{array}\right]\left[\begin{array}{c} 0\\ 0\\ -g \end{array}\right]$$

Then,
(19)$$C_{3}=\left[\begin{array}{ccc} f_{sfx}^{b}/g & f_{sfy}^{b}/g & f_{sfz}^{b}/g \end{array}\right]$$

Equation (Equation 19) represents the third row of the attitude matrix $C_{b}^{n}$. The elements of the first two rows can be chosen arbitrarily, provided that they satisfy the right-hand rule.

### 2.2. Human Step Length Model

As the error of the pedestrian navigation system accumulates over time, it is feasible to use the motion constraints of pedestrians as error constraints. Based on this, more accurately grasping the kinematic characteristics of pedestrians by establishing a human step length model is one of the key means of improving the positioning accuracy and reliability of autonomous navigation systems.

#### 2.2.1. Data Collection

Data collection is performed following time synchronization and system calibration, utilizing both the Optitrack optical motion capture system and the FOHEART MAGIC inertial motion capture system.

Table 1 shows the data collection program, which involved selecting 10 volunteers to complete the human kinematics data collection. The “Load” refers to the weight a person carries when moving. The data collection modes in this case include walking on a treadmill at speeds of 1, 2, 2.5, 3, 3.5, 4, 4.5, and 5 km/h for one minute each, as well as free-walking and free-jogging for three laps each, resulting in a total of ten modes.

#### 2.2.2. Data Processing

The process of data collection involves five stages: real-time data collection, data pre-processing, conversion to standard data formats, data slicing, and database storage. The human kinematics data collection site is shown in Figure 1.

By calculating the distance between the two feet, the variation pattern of the step length under different traveling speeds and weight loads can be obtained, and then fitted with a linear function:(20)StepLength=2.789×Height−3.897

### 2.3. Bipedal Constraint Algorithm Based on Step Length Model

The ZUPT algorithm can effectively limit the correction of velocity state quantities in the navigation system. However, it does not provide sufficient constraints on the position and heading angles. This paper presents a bipedal constraint algorithm based on the step length model for two IMUs located in each foot. The algorithm implements correction constraints on the heading angle and position to enhance the system’s navigation accuracy.

According to the human step length model, there is a maximum distance threshold between the dual feet of a pedestrian while walking. When one of the feet is in a stationary state and the other foot is in a swinging state, if the stationary foot is used as the coordinate origin, the swinging foot is within a ball with the origin as the center and the threshold as the radius. If the distance between the two feet exceeds the threshold range, a virtual position observation can be constructed based on the distance between the feet and the threshold. This observation can then be combined with the Kalman filter to correct the navigation system.

In SINS, the position *p* consists of the longitude *L*, latitude λ, and height *h*. When dealing with the relationship between two IMUs, it is more intuitive and convenient to use the navigation coordinate system based on east, north, and sky. The conversion formula is as follows:(21)$$\left\{\begin{array}{cc} P_{E} & =\sin L(R_{N}+h)\\ P_{N} & =\lambda (R_{M}+h)\\ P_{U} & =h \end{array}\right.$$

The position coordinates are $P_{E}$ for the east, $P_{N}$ for the north, and $P_{U}$ for the sky. Also, noting that the spacing between two IMUs is *D*, here is
(22)$$\begin{array}{cc} D & =\left|P_{r}-P_{l}\right|={\left[{\left(P_{E,r}-P_{E,l}\right)}^{2}+{\left(P_{N,r}-P_{N,l}\right)}^{2}+{\left(P_{U,r}-P_{U,l}\right)}^{2}\right]}^{1/2} \end{array}$$

Here, *l* and *r* represent the left foot IMU and the right foot IMU, respectively. $P_{r}$ and $P_{l}$ represent the positions of the left foot IMU and the right foot IMU, respectively. Assume that when the left foot is in a stationary state and the right foot is in a swinging state, if the distance between the two feet is greater than the set threshold *B*, then
(23)$$\begin{array}{ccc} {\hat{P}}_{r} & =P_{l}+\frac{B}{D}\left(P_{r}-P_{l}\right)\; & =\frac{1}{D}\left(\left(D-B\right)P_{l}+BP_{r}\right) \end{array}$$
(24)$$R_{r}=\sigma_{p}^{2}I_{3}$$

Here, ${\hat{P}}_{r}$ is the virtual observation position of the right foot position at this time, $\sigma_{p}$ represents the variance of position, *I* represents the identity matrix, and $R_{r}$ is the covariance matrix. After converting the resulting east, north, and sky positions back to longitude, latitude, and altitude via Equation (Equation 21), the virtual observations when using the cooperative navigation algorithm are obtained:(25)$$Z_{\mathrm{Bipedal}}=\left[\begin{array}{c} L-\hat{L}\\ \lambda -\hat{\lambda }\\ h-\hat{h} \end{array}\right]$$

And the corresponding observation matrix is
(26)$$H_{\mathrm{Bipedal}}=\left[\begin{array}{ccc} 0_{3\times 6} & I_{3} & 0_{3\times 6} \end{array}\right]$$

To provides an overview of the navigation algorithm. The pseudocode is listed in Algorithm 1.

When a pedestrian walks, their feet alternate between contacting the ground, resulting in a zero-speed state. To determine when both feet are at zero speed, a zero-speed determination is performed, and the ZUPT algorithm is applied to the feet in this state. Additionally, when one foot is stationary and the other is swinging, the system makes a threshold judgment on the distance between the two feet. If the swinging foot exceeds the range of the stationary foot, cooperative navigation constraints are applied to the swinging foot. The system also includes a barometer to provide altitude information for the navigation system to use as a constraint. The altitude constraint is performed in the same way as the zero-speed correction algorithm, that is, when the foot is at zero-speed. In this paper, the bipedal spacing constraint threshold is provided by the human step length model.
**Algorithm 1** Bipedal Constraint Algorithm Based on Step Length Model.**Require:** Δv and Δθ measured by bipedal IMUs**Ensure:** Attitude, speed, and position of bipedal navigation systems k←0 SINS_*l*_
← Initial system state and Kalman filter of left foot SINS_*r*_
← Initial system state and Kalman filter of right foot **loop**  k←k+1  **if**
$\left(Stance_{l}=1\right)$ or $\left(Stance_{r}=1\right)$
**then**   Perform the ZUPT algorithm on the foot at zero speed   $Z_{\mathrm{ZUPT}}={\left[\begin{array}{ccc} V_{E} & V_{N} & V_{U} \end{array}\right]}^{T}$   $H_{\mathrm{ZUPT}}=\left[\begin{array}{ccc} 0_{3\times 3} & I_{3\times 3} & 0_{3\times 9} \end{array}\right]$   Constrains $h_{Stance=1}$ using the barometer  **end if**  **if**
(Stance=1)⊕(Stance=1)=1
**then**   $D=\left|P_{r}-P_{l}\right|$   **if**
D>B
**then**    Algorithmically constrains feet in non-zero speed states    ${\hat{P}}_{r}=P_{l}+\frac{B}{D}\left(P_{r}-P_{l}\right)=\frac{1}{D}\left(\left(D-B\right)P_{l}+BP_{r}\right)$    $Z_{\mathrm{Bipedal}}={\left[\begin{array}{ccc} L-\hat{L} & \lambda -\hat{\lambda } & h-\hat{h} \end{array}\right]}^{T}$    $H_{\mathrm{Bipedal}}=\left[\begin{array}{ccc} 0_{3\times 6} & I_{3} & 0_{3\times 6} \end{array}\right]$   **end if**  **end if** **end loop**

## 3. Experiment and Results

The effectiveness of the proposed method was confirmed through experiments. The experiment utilized a self-developed pedestrian navigation module with TI’s CC2652P1FRGZR as the main processor and communication processor. The LPD22HBTR collects air pressure data, while the six-axis IMU sensor ASM330LLH collects acceleration and angular velocity data. Additionally, the three-axis geomagnetic sensor MMC5883 is used to collect geomagnetic data. The collected data are encoded in hexadecimal and transmitted to the tablet through the communication module for storage. The parameters of the sensors are shown in Table 2. The pedestrian module can be powered by either a USB port or a battery. Figure 2 shows the foot module prototype, which controls the starting and stopping of the system through a switch.

During the experiment, the navigation modules of both feet were powered on and connected to the tablet. The navigation modules were placed side by side on the ground for 10 s to preheat and initialize, and then installed on the left and right feet, respectively. The experimenter completed two walks in different locations, labeled Walk 1 and Walk 2 (see Figure 3 and Figure 4, respectively), returned to the starting point, stopped collecting data, and recorded the test results.

The experimenter was 24 years old, 183 cm tall, weighed 70 kg, and was not weight-bearing. The maximum stride length during normal walking was obtained using the human step length model. The distance threshold between the two IMUs in the bipedal constraint algorithm was set to *B* = 1.2 m. Figure 5, Figure 6 and Figure 7 show the result of Walk 1, and Figure 8, Figure 9 and Figure 10 show the result of Walk 2. Walk 1 was undertaken in the corridor on the second floor of Tsinghua University’s FIT building, as shown in Figure 3. Walk 2 was undertaken outside Tsinghua University’s Weiqing building, as shown in Figure 4.

Analyzing the results of the experiment, it can be seen that, although ZUPT alone can be used for speed constraints and preliminary navigation solutions can be performed, it cannot effectively constrain the heading angle error during the navigation process. As the navigation time increases, the trajectories will gradually diverge, and the navigation settlement results of the two feet will gradually expand from each other. This is obvious when walking in a straight line. As the pedestrian walks in a straight line, the navigation results of the two feet gradually deviate from the original heading, and the distance between them gradually increases. When there are turns during walking, the navigation errors accumulated due to heading errors will be more significant, and this error will increase with the increase in turns. In the end, it is impossible to return to the starting point of the trip relying solely on the navigation solution results of ZUPT. After adopting the bipedal constraint algorithm, the INS on the left and right sides provide each other with position observations, and the Kalman filter effectively reduces the errors in the navigation results of the left and right INS due to gyroscope drift. Therefore, it is necessary to introduce the bipedal constraint algorithm to constrain the position and heading angles.

The radial error results of the navigation solutions before and after the introduction of the bipedal constraint algorithm are shown in Table 3.

The left and right biped radial errors of Walk 1 are reduced by 53.07% and 86.08%, respectively, the left and right biped radial errors of Walk 2 are reduced by 60.84% and 45.46%, respectively, and the mean biped radial errors of Walk 1 and Walk 2 are reduced by 68.16% and 50.61%, respectively. And Table 4 shows a comparison of the proposed algorithm with other algorithms. It can be seen that, by introducing the bipedal constraint algorithm, the positioning accuracy of the navigation algorithm has been significantly improved, and our proposed algorithm has better results.

## 4. Conclusions

In this article, we used motion capture technology to collect human kinematic data from multiple volunteers, and a human step model was established based on this. Secondly, given the defect that the ZUPT algorithm cannot effectively constrain the heading angle, we proposed the bipedal constraint algorithm based on the established human step length model, which was used to construct a virtual observation of the relative position. Real field experiments demonstrated that, by introducing the bipedal constraint algorithm, the mean biped radial errors of Walk 1 and Walk 2 were reduced by 68.16% and 50.61%, respectively. The results of the experiments indicate that the algorithm effectively reduces the radial error of the navigation results and improves the navigation accuracy.

In future work, more types of human kinematics data of gaits can be collected to obtain human kinematics conditions under different gaits, and then different constraint algorithms can be proposed for different gaits to further improve the accuracy of the navigation system. Secondly, the algorithm proposed in this article, combined with the ZUPT algorithm, effectively suppress the divergence of navigation errors, but do not completely eliminate it. Considering that the heuristic course correction algorithm also has an effective constraint effect on the course angle, combining the algorithm in this paper with it may lead to better navigation results. In addition, the current algorithm only applies to two feet. In the future, we should consider adding new nodes, such as at the waist and head, to further improve the robustness of the algorithm.

## Figures and Tables

**Figure 1 sensors-24-01073-f001:**
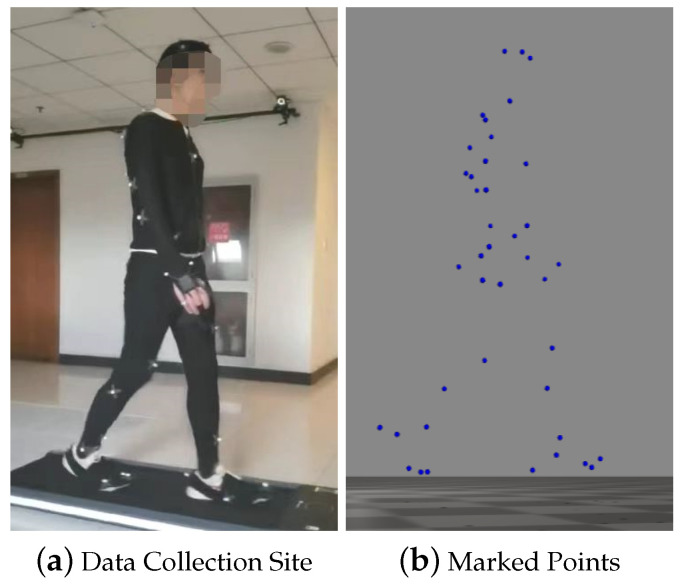
Data collection site and marked points: (**a**) shows the data collection site and (**b**) shows the marked points.

**Figure 2 sensors-24-01073-f002:**
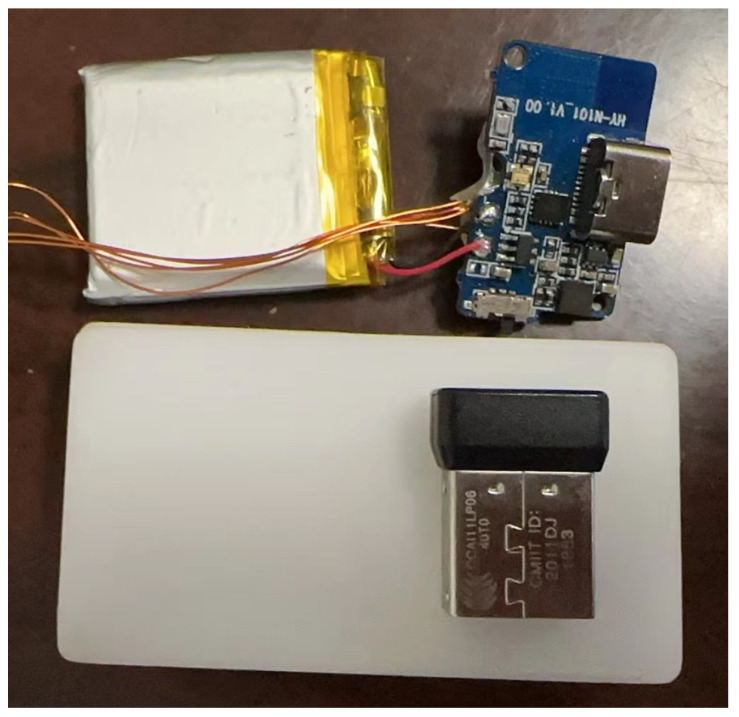
Navigation module prototype.

**Figure 3 sensors-24-01073-f003:**
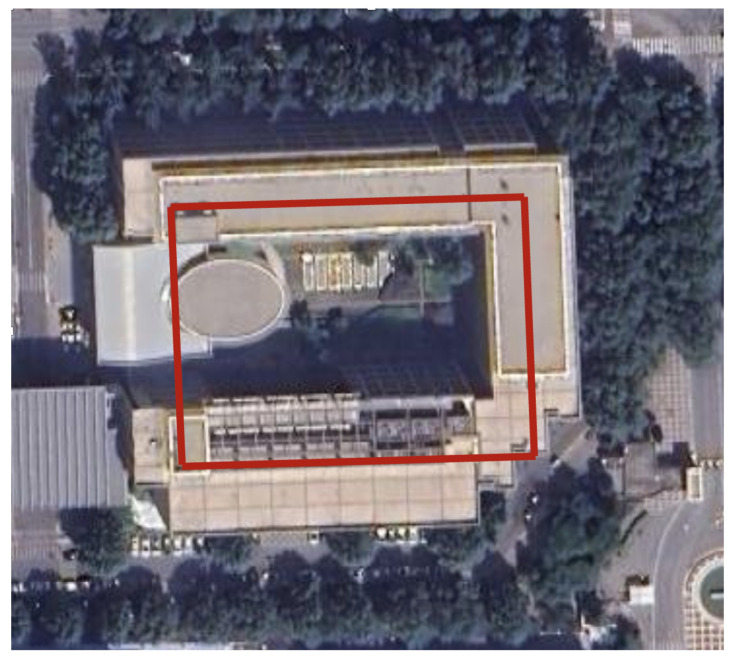
Tsinghua University’s FIT building (Walk 1).

**Figure 4 sensors-24-01073-f004:**
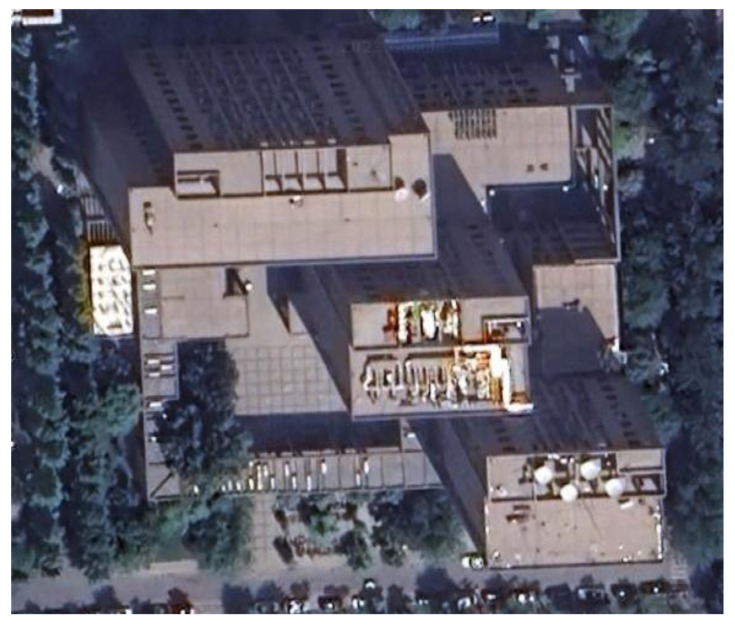
Tsinghua University’s Weiqing building (Walk 2).

**Figure 5 sensors-24-01073-f005:**
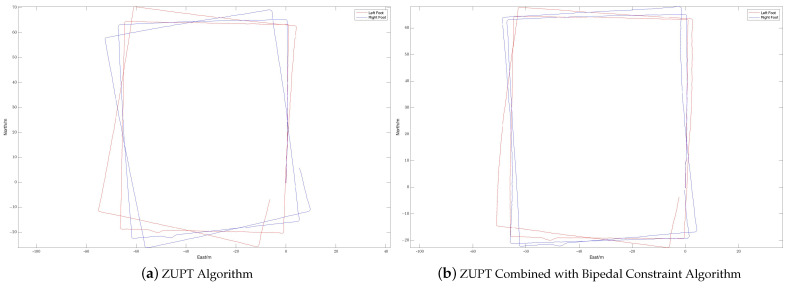
Displacement trajectory of Walk 1: (**a**) is the displacement trajectory with only the ZUPT algorithm and (**b**) is the displacement trajectory with ZUPT combined with the bipedal constraint algorithm.

**Figure 6 sensors-24-01073-f006:**
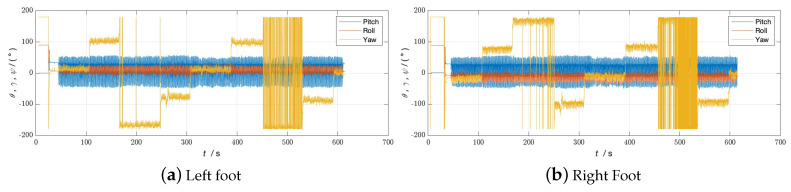
Attitude angle of Walk 1 without bipedal constraint algorithm.

**Figure 7 sensors-24-01073-f007:**
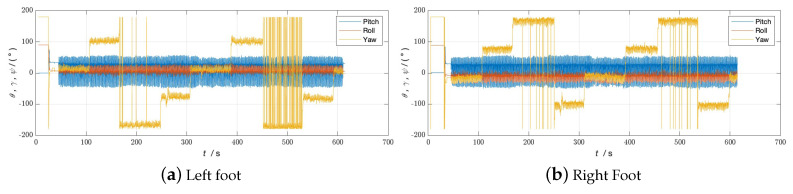
Attitude Angle of Walk 1 with bipedal constraint Algorithm.

**Figure 8 sensors-24-01073-f008:**
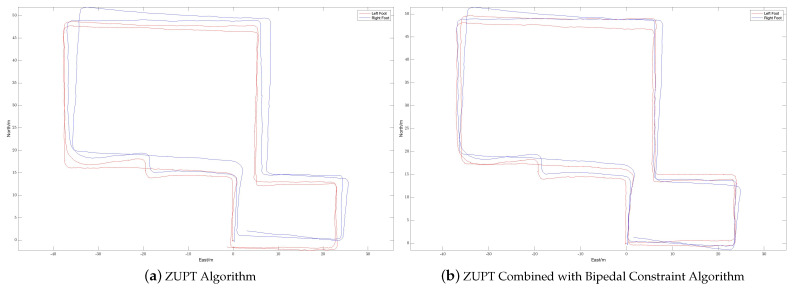
Displacement trajectory of Walk 2: (**a**) is the displacement trajectory with only the ZUPT algorithm and (**b**) is the displacement trajectory with ZUPT combined with the bipedal constraint algorithm.

**Figure 9 sensors-24-01073-f009:**
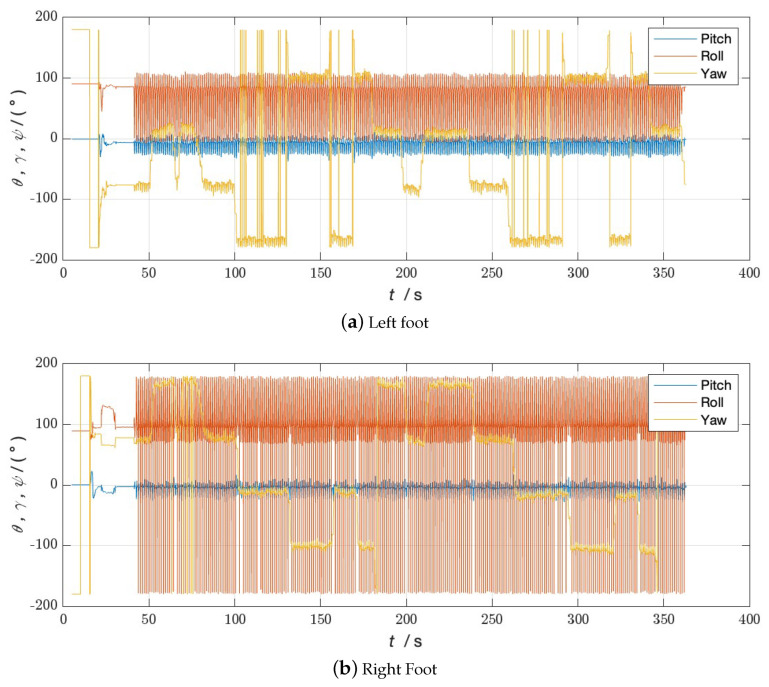
Attitude angle of Walk 2 without bipedal constraint algorithm.

**Figure 10 sensors-24-01073-f010:**
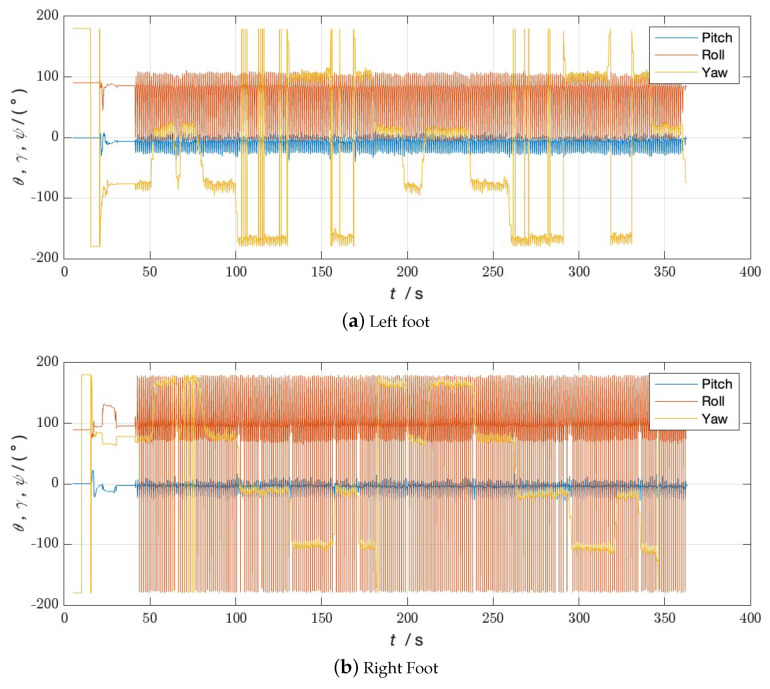
Attitude angle of Walk 2 with bipedal constraint algorithm.

**Table 1 sensors-24-01073-t001:** Data collection program.

No.	Sex	Height/cm	Load/kg
1	Male	165	0, 10, 15
2	Male	170	0, 10, 15
3	Male	175	0, 10, 15
4	Male	180	0, 10, 15
5	Male	185	0, 10, 15
6	Male	190	0, 10, 15
7	Female	158	0, 5
8	Female	165	0, 5
9	Female	170	0, 5
10	Female	175	0, 5

**Table 2 sensors-24-01073-t002:** Specification of the sensors.

Specification	Barometer	Accelerometer	Gyroscope
Full scale	1260 hpa	±16 g	±4000 dps
Bias stability	-	-	3 deg/h
Noise	0.75 $\mathrm{pa}/\sqrt{\mathrm{Hz}}$	60 $\mu g/\sqrt{\mathrm{Hz}}$	5 $\mathrm{mdps}/\sqrt{\mathrm{Hz}}$
Temperature	−40∼+85 °C	−40∼+105 °C	

**Table 3 sensors-24-01073-t003:** Experimental results.

Experiment	Radial Error without Algorithm (m)			Radial Error with Algorithm (m)			Improvement		
	Left Foot	Right Foot	Mean	Left Foot	Right Foot	Mean	Left Foot	Right Foot	Mean
Walk 1	9.3999	7.9166	8.6583	4.4108	1.1022	2.7565	53.07%	86.08%	68.16%
Walk 2	1.8467	3.6703	2.7585	0.7231	2.0017	1.3624	60.84%	45.46%	50.61%

**Table 4 sensors-24-01073-t004:** Comparison of four algorithms in terms of position error.

Algorithm	Proposed by Xu [28,29]	Proposed by Shi [26,27]	Proposed by Niu [31,32]	Proposed in This Paper
Improvement	About 30%	41.46%	About 50%	68.16%/50.61%

## Data Availability

The raw data supporting the conclusions of this article will be made available by the authors on request.

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
