# Peer review of "Data Fusion of Dual Foot-Mounted INS Based on Human Step Length Model"

_sensors, 2024, doi:10.3390/s24041073_

Round 1

Reviewer 1 Report

Comments and Suggestions for Authors

1- Abstract should be rewritten to better describe the article. If you use abbreviations in the abstract, you should also give the spelling or do not use abbreviations.

2 - The sentences in the article are too long in some parts, making it difficult to understand. For example, line 20-24. The sentence structures of the article should be reorganized.

3 - The contributions of the article to the literature should be written in detail at the end of the introduction.

4- The algorithms used are described in sufficient detail.

5- The experimental study and experimental results are adequately presented.

6- The experimental results are presented and discussed in detail. However, the results obtained should be compared with the results of studies in the literature.

7- The numerical results obtained from the experimental study should be given in the conclusion and abstract sections.

Reviewer 2 Report

Comments and Suggestions for Authors

1.       The limiting distance between the feet is very important to the algorithm of this paper, how is D=1.2 m given in line 228 calculated?

2.       Why are the difference of yaw for the left and right feet so great in Figures 9 and 10? How does this affect the algorithm in this paper?

3.       Table 3 gives cumulative error results that are strongly correlated with the total length of the route traveled and do not represent the true performance of the algorithm. It is recommended that the cumulative error per kilometer and the improvement rate be used as a measure.

4.       From the experimental results, there is a large difference between the trajectory data of the left and right feet. How should these two sets of data be used to build the final real route in practical applications?

Reviewer 3 Report

Comments and Suggestions for Authors

The article is devoted to the development of the human step length model based on motion capture technology.  To reduce the heading angle error during the navigation process, authors propose the bipedal constraint algorithm. Authors compare their method with ZUPT algorithm and report significant improvement.

The topic of the article is relevant. The text is well structured and well written.

Some comments:

1) Line 69 – Abbreviation MIMU is not described.

2) The Introduction section should contain a brief statement of the research gap, the objectives of the study and the main contributions of the work. Now it's completely missing. Since the authors do not formulate the purpose of the work, it is not possible to assess the degree to which the research goal has been achieved.

3) At the end of the Introduction it is necessary to inform the reader about the structure of the article.

4) In expressions (1)-(6), (13)-(15),(21),(22),(24) some components are not described.

5) Formulas must have punctuation.

6) In Table 1, the meaning of column “Load” is unclear.

7) Figures 3 and 4, Table 3 must be mentioned in the text before they appear.

8) Walk1 and Walk2 are not labeled in the Figure 3 and Figure 4, as it described in the text. Besides, when Walk1 and Walk2 are mentioned in the abstract, it is not clear to the reader what the authors mean by them.

9) Reference 26 is formatted incorrectly.

Comments on the Quality of English Language

In general, English is at a sufficient level, but there are some minor imperfections like

- different spelling (“program” and “programme”, “Walk 1” and “walk 1”)

- “Figure 5 – 7” instead of “Figures 5-7”

- typos (“Zero Velocity Updat”)

and so on.

Round 2

Reviewer 1 Report

Comments and Suggestions for Authors

Thanks for your efforts to make the article easier to read.

The authors have carefully made the edits I requested in the review report. There is no harm in publishing the manuscript in this form.